# The Mean Staple Length of Wool Fibre Is Associated with Variation in the Ovine Keratin-Associated Protein 21-2 Gene

**DOI:** 10.3390/genes11020148

**Published:** 2020-01-30

**Authors:** Shaobin Li, Huitong Zhou, Hua Gong, Fangfang Zhao, Jiqing Wang, Xiu Liu, Jiang Hu, Yuzhu Luo, Jon G.H. Hickford

**Affiliations:** 1Faculty of Animal Science and Technology, Gansu Key Laboratory of Herbivorous Animal Biotechnology, Gansu Agricultural University, Lanzhou 730070, China; Lisb@gsau.edu.cn (S.L.); wangjq@gsau.edu.cn (J.W.); liuxiu@gsau.edu.cn (X.L.); huj@gsau.edu.cn (J.H.); 2International Wool Research Institute, Gansu Agricultural University, Lanzhou 730070, China; Huitong.Zhou@lincoln.ac.nz (H.Z.); Hua.gong@lincoln.ac.nz (H.G.); 3Gene-marker Laboratory, Faculty of Agricultural and Life Sciences, Lincoln University, Lincoln 7647, New Zealand

**Keywords:** KAP21-2 gene (*KRTAP21-2*), mean staple length, wool traits, sheep

## Abstract

Wool and hair fibres consist of a variety of proteins, including the keratin-associated proteins (KAPs). In this study, a putative ovine homologue of the human KAP21-2 gene (*KRTAP21-2*) was identified. It was located on chromosome 1 as a 201-bp open reading frame (ORF) in the ovine genome assembly from a Texel sheep (v.4 NC_019458.2: nt122932727 to 122932927). A polymerase chain reaction- single strand conformation polymorphism (PCR-SSCP) analysis of this ORF, and subsequent DNA sequencing, identified five sequences (named *A*-*E*). The putative amino acid sequences that would be produced, shared some identity with each other and with other KAPs, but they were most similar to ovine KAP21-1, and phylogenetically related to human KAP21-2. The location of the ovine *KRTAP21-2* sequence was consistent with the location of human *KRTAP21-2,* and this suggests they represent different variant forms of ovine *KRTAP21*-2. Variation in this gene was investigated in 389 Merino (sire) × Southdown-cross (ewe) lambs. These were derived from four independent sire-lines. The sequence variation was found to be associated with variation in five wool traits: including mean staple length (MSL), mean fibre diameter (MFD), fibre diameter standard deviation (FDSD), prickle factor (PF), and greasy fleece weight (GFW). The most persistent effect of *KRTAP21-2* variation was with variation in MSL; with the MSL of sheep of genotype *AC* being 12.5% greater than those of genotype *CE*. A similar effect was observed from individual variant absence/presence models. This suggests that *KRTAP21-2* should be further investigated as a possible gene-marker for improving MSL.

## 1. Introduction

Wool is a complex structure composed of numerous proteins. Of these, the large and diverse family of keratin-associated proteins (KAPs) fulfil an important structural role, as they are components of the matrix in which the keratin intermediate filaments (KIFs) are embedded. They are believed to play an important role in defining the physical and mechanical properties of wool fibres [1]. The KAPs usually possess either a high level of cysteine, or both glycine and tyrosine, and historically they have been classified into three groups based on their amino acid content: the ultra-high sulphur (UHS) KAPs, the high sulphur (HS) KAPs, and the high glycine-tyrosine (HGT) KAPs [2].

Of these KAPs, HGT-KAPs are predominantly found in the wool fibre orthocortex, and they have been revealed to be the first KAPs expressed after the synthesis of the KIFs. The proportion of the wool fibre that is HGT-KAP varies in different wools, varying up to 12% by weight in the wool of Merino sheep, to less than 1% by weight in wool from Lincoln sheep [3]. The small amount of HGT-KAP in sheep with the felting lustre mutation and the wide range of content of HGT-KAP in wool from different sheep breeds [4], suggests these proteins have a novel role in determining wool fibre characteristics.

Seventeen HGT-KAP genes have been described in humans [5], but only twelve HGT-KAP genes that belong to six KAP families (KAP6–KAP8 and KAP20–KAP22) including a recently described KAP21-1 gene, have been identified in sheep [2,6,7,8,9]. While there are two KAP21 genes (*KRTAP21-1* and *KRTAP21-2*) in humans [10], only *KRTAP21-1* has been described in sheep [9], which suggests the possible existence of another ovine KAP21 gene. 

To confirm this, we analysed the sheep genome assembly sequence (NC_019458.2) in proximity to ovine *KRTAP21-1* [9] and at approximately 4.6 kb downstream of the *KRTAP21-1* identified an open reading frame (ORF) (NC_019458.2: 122932727 to 122932927) that could potentially produce a putative HGT-KAP (unpublished data). In this study, we describe the characterization of this ORF, report sequence variation in it that was detected by polymerase chain reaction-single strand conformation polymorphism (PCR-SSCP) analysis and DNA sequencing, and reveal associations between this gene variation and variation in some important wool fibre traits.

## 2. Materials and Methods

The collection of drops of blood from the ears of sheep as was undertaken in this study, is covered by Section 7.5 Animal Identification of the Animal Welfare (Sheep and Beef Cattle) Code of Welfare 2010. This code of welfare is issued under the New Zealand Animal Welfare Act 1999 (New Zealand Government).

### 2.1. Sheep Blood and Wool Samples

Three hundred and eighty-nine Merino × Southdown-cross lambs from four sire-lines farmed at Ashley Dene, Canterbury, New Zealand (NZ), and NZ Romney lambs (n = 75) from five separate farms, were used to ascertain if there was DNA sequence variation in the putative *KRTAP21-2*. The 389 Merino × Southdown-cross lambs were also used in the association analysis. Blood samples from each of the lambs was collected onto individual blotting papers (FTA^TM^ cards, Whatman BioScience, Middlesex, UK) and genomic DNA from the leucocytes in the blood was prepared using the method described by Zhou et al. [11].

The Merino × Southdown-cross lambs were all ear-tagged with an identification number within 12 hours of parturition and their birth date, birth weight, birth rank (i.e. whether they were a single, twin or triplet), gender, and pedigree (sire and ewe identity) were recorded. The ewes and lambs remained as a single mob until weaning, at which time the lambs were separated from the ewes and the lambs were separated into two groups based on their gender. At approximately one year of age the lambs were shorn, and their greasy fleece weight (GFW) was measured. Mid-side wool samples were collected and sent for wool testing. This was undertaken by the New Zealand Wool Testing Authority Ltd (Ahuriri, Napier, NZ) using International Wool Textile Organisation (IWTO) endorsed methods (www.iwto.org). The traits measured at testing included mean staple length (MSL), mean fibre curvature (MFC), and mean staple strength (MSS); plus the fibre diameter related traits of mean fibre diameter (MFD), fibre diameter standard deviation (FDSD), coefficient of variation of fibre diameter (CVFD), and prickle factor (PF). Wool yield (Yield) was assessed and this was used to calculate the clean fleece weight (CFW).

### 2.2. PCR Primers and Amplification

Sequences bordering the newly identified ORF (NC_019458.2: 122932727 and 122932927) were analysed to assist in the design of PCR primers that would span the entire putative HGT-KAP reading frame. The chosen primer sequences were: 5’-ACACACTTCAGAACCATCGC-3’ and 5’-TGGTTTC AGACGTAAATGGTG-3’, and they were made by Integrated DNA Technologies (Coralville, IA, USA). The PCR amplifications were performed in a 15-μL reaction. Each reaction contained the genomic DNA on an individual 1.2-mm punch of a single FTA paper, 0.5 U of Taq DNA polymerase (Qiagen, Hilden, Germany), 1 × reaction buffer supplied with the enzyme, 0.25 μM of each primer, 150 μM of each dNTP (Bioline, London, UK), and 2.5 mM of Mg^2+^. Amplification was carried out in S1000 cyclers (Bio-Rad, Hercules, CA, USA), with a thermal profile that consisted of two minutes at 94 °C, followed by 35 cycles of 30 s at 94 °C, 30 s at 62 °C and 30 s at 72 °C, with a final incubation of five minutes at 72 °C. 

### 2.3. Screening for Variation in KRTAP21-2

The PCR amplicons were screened for DNA sequence variation using a SSCP analysis. For each amplicon derived from PCR, a 0.7-μL aliquot was added to 7 μL of 98% formamide, 10 mM EDTA, 0.025% bromophenol blue, and 0.025% xylene-cyanol. These were then denatured for five minutes at 95 °C for five minutes. The tubes containing the amplicons and dye were then cooled in wet ice and immediately loaded into separate lanes on 16 cm × 18 cm, 14% acrylamide: bisacrylamide (37.5:1) (Bio-Rad) gels. These gels contained 3% v/v glycerol. The SSCP run entailed electrophoresis at 350 volts in Protean II xi cells (Bio-Rad) with a 0.5× TBE running buffer. Buffer temperature was precisely maintained at 19 °C for the full 24 h of electrophoresis. After electrophoresis, the SSCP gels were silver-stained [12].

### 2.4. Sequencing of PCR-SSCP Variants and Sequence Analysis

If the amplicons produced PCR-SSCP banding patterns that suggested the sheep was homozygous in the amplified region, then they were sequenced in both directions at the Lincoln University (New Zealand) DNA sequencing facility. If the PCR-SSCP banding patterns were more complex and the patterns suggested the sheep were heterozygous for the amplified region, then a band corresponding to a single amplicon/ variant was excised as a gel slice. This was macerated then used as a template for re-amplification with the original primers. This approach is described by Gong et al. [13]. The translation of DNA sequences to putative amino acid sequences and other sequence alignments were undertaken using DNAMAN (version 5.2.10, Lynnon BioSoft, Vaudreuil, QC, Canada). 

### 2.5. Statistical Analyses

The data were analysed statistically using Minitab version 16 (Minitab Inc., State College, PA, USA). In the analyses, General Linear Models (GLMs) were employed to assess the effect of the presence or absence of the different *KRTAP21-2* variants on variation in the measured wool traits. The approach involved starting with single-variant models, where only a single variant’s presence/absence was factored into the models. In these models, any variant that was revealed to associate with variation in a wool trait with a low threshold for rejection of the null hypothesis (*p* < 0.2), and which could this potentially be affecting the trait, was then factored into a second series of multi-variant models. In effect the presence/absence of any given allele was therefore corrected for the effect of any other alleles in the sheep’s genotype that might being affecting the traits. 

Next GLMs were used to compare the wool traits between lambs that had different *KRTAP21-2* genotypes. This was only done using genotypes with a frequency greater than 5%, and so as to limit the potential introduction of bias from small groups of sheep with less common genotypes. When the GLMs indicated significant differences among the genotypes, multiple pairwise comparisons of the sheep with the different genotypes were made, and with a correction (Bonferroni) being applied to reduce the chances of obtaining false-positive results during the multiple comparisons. 

In the GLMs, the sire of the lambs was revealed to affect all the wool traits. Accordingly, sire was included as a random explanatory factor in all the models. The gender of the lambs was revealed to have an effect on GFW, CFW, MSL, MFD, FSDS, MSS, MFC and PF, and hence gender was included as a fixed explanatory factor in the models for these traits. The birth rank of the lambs had an effect on MSL, and hence it was included as a fixed explanatory factor in the model for MSL. 

## 3. Results

### 3.1. Identification of KRTAP21-2 in the Sheep Genome

A 201-bp ORF was identified between OAR1: NC_019458.2, 122932727 and 122932927. Eighteen other KAP genes have also been identified in this vicinity and in order from the centromere to telomere these were *KRTAP24-1*, *KRTAP28-1*, *KRTAP26-1*, *KRTAP13-3*, *KRTAP15-1*, *KRTAP20-1*, *KRTAP6-3*, *KRTAP22-1*, *KRTAP6-1*, *KRTAP6-4*, *KRTAP6-2*, *KRTAP6-5*, *KRTAP20-2*, *KRTAP21-1*, *KRTAP8-2*, *KRTAP8-1*, *KRTAP7-1* and *KRTAP11-1* (Figure 1). The PCR primers were designed to amplify the ORF and upon sequencing the amplicons were confirmed to be of 294 bp in size. Following sequencing, the ORF would potentially produce a HGT-KAP that was most similar at the level of amino acid sequence to ovine KAP21-1 (Figure 2), and phylogenetically related to human KAP21-2. This suggests that this ORF represents the ovine KAP21-2 gene and hence it was named ‘SHEEP-*KRTAP21-2*’, using the updated KAP/*KRTAP* nomenclature [14].

### 3.2. Variation in SHEEP-KRTAP21-2

There were five PCR-SSCP banding patterns detected for the amplicon containing the ORF, with either one or a combination of two banding patterns being observed for each sheep (Figure 3). DNA sequencing of the amplicons revealed five nucleotide sequences (named *A*, *B*, *C*, *D* and *E*), and these sequences were deposited into GenBank with accession numbers MF143975–MF143979. Four nucleotide sequence differences were identified when comparing the five sequences (Table 1). All of these were located in the ORF, and if the ORF was functional then one of them would be a non-synonymous substitution that would result in an amino acid change (p.Val56Ile) in the putative KAP21-2 protein. 

The putative *KRTAP21-2* sequences would, if expressed, encode a 66 amino acid polypeptide. This polypeptide would contain a high content of glycine (22.73 mol%) and tyrosine (21.21 mol%), and moderate levels of cysteine (16.67 mol%), serine (12.12 mol%), arginine (6.06 mol%), and asparagine (6.06 mol%). Other residues that were commonly found in the polypeptide included proline (3.03 mol%) and eight other amino acids (aspartic acid, glutamic acid, histidine, leucine, methionine, phenylalanine, threonine, valine) accounted for 1.52 mol%. The putative KAP21-2 protein would therefore appear to be a basic protein, with a pI value (predicted) of approximately 7.8. 

### 3.3. Comparison of the Variant and Genotype Frequencies in NZ Romney and Merino × Southdown-Cross Sheep 

The frequencies of the *KRTAP21-2* variants in the NZ Romney sheep were: *A*: 6.85%; *B*: 19.86%; *C*: 53.42% and *E*: 19.86%; while those in Merino × Southdown-cross sheep were: *A*: 21.36%, *B*: 30.56%, *C*: 35.81%, *D*: 2.3%, and *E*: 9.97%. Variant *C* was common in the NZ Romney sheep, while in the Merino × Southdown-cross sheep, both *C* and *B* were common. Variant *A* was common in the Merino × Southdown-cross sheep, but was rarer in the NZ Romney sheep. The frequency of *E* was much higher in the NZ Romney sheep than the Merino × Southdown-cross sheep. 

Ten genotypes were distinguished in the Merino × Southdown-cross sheep, and they were as follows: *AA*, *AB*, *AC*, *AE*, *BB*, *BC*, *BD, BE*, *CC*, *CD*, *CE* and *EE*, while *BD* and *CD* were not found in the Romney sheep. Of the genotypes present in the Merino × Southdown-cross sheep, only seven (*AB*, *AC*, *BB*, *BC*, *BE*, *CC* and *CE*) occurred at a frequency over 5%, and accordingly only sheep of these genotypes were used in the genotype models.

### 3.4. Effect of Variation in SHEEP-KRTAP21-2 on Wool Traits

Of the five *KRTAP21-2* variants detected in Merino × Southdown-cross lambs, *D* occurred at a very low frequency, and so its association with wool traits was not tested. In the ‘single-variant’ models, the presence of *E* was associated with a decrease in MSL (*p* = 0.003), and the effect of *E* on MSL persisted when *A* was introduced into models (*p* = 0.013). Variant *E* was also trending towards an association with lower GFW (*p* = 0.173), and increased MFD (*p* = 0.102), FDSD (*p* = 0.068), MSS (*p* = 0.068), and PF (*p* = 0.055) (Table 2). The presence of *A* was associated with an increase in GFW (*p* = 0.030), but the association between *A* and GFW was lost when *E* was introduced into the model. A trend for association was still evident (*p* = 0.078).

### 3.5. Effect of Common Genotypes on Wool Traits 

With the seven common *KRTAP21-2* genotypes (*AB*, *AC*, *BB*, *BC*, *BE*, *CC* and *CE*), an effect of genotype was observed for MSL (*p* = 0.026), FDSD (*p* = 0.019), PF (*p* = 0.041), and a trend towards association was observed for GFW (*p* = 0.054) and MFD (*p* = 0.060) (Table 3). Sheep of genotype *AC* were of a predicted mean MSL that was 12.5% greater than those of genotype *CE*. Genotype *CE* was associated with a higher predicted mean FDSD, than *CC* and *BC*. In terms of PF, sheep of genotype *CE* had a predicated mean PF of 4.72%, this being more than 2.3 times higher than that of sheep with *BC* and *CC*.

## 4. Discussion

The putative SHEEP-*KRTAP21-2* was located on chromosome 1 and was clustered with other KAP genes that have been reported previously. It displayed a lower similarity to any previously described ovine HGT-KAP gene, but was closer in identity to a human *KRTAP21-2* sequence. This gene was also located between *KRTAP20-2* and *KRTAP21-1*, and this location is consistent with the chromosomal location of human *KRTAP21-2* [10].

If transcribed and translated the amino acid sequence that would be produced from the *KRTAP21-2* sequence would contain 66 residues and nearly half of these would be glycine or tyrosine. This is consistent with the defining characteristic of the HGT-KAPs. However, this polypeptide would also contain a high level (16.67 mol%) of cysteine, which is more than has been reported for any other HGT-KAP family. It is comparable to that reported for other ovine HS-KAPs though. This higher than expected level of cysteine had also been described for the other SHEEP-KAP21 family member, KAP21-1 [9]. This suggests that while KAP21-2 can be classified as a HGT-KAP, it could also be called a HS-KAP; and functionally by way of the formation of disulphide bridges from the cysteine residues, form cross-links with other HS-KAPs and the keratin intermediate filaments in the wool fibre. Additionally, it would suggest the current broad classification of KAPs into three groups (HS-, UHS- and HGT-KAPs), may need to be revised.

Four nucleotide sequence variations were found in ovine *KRTAP21-2* and these produced five unique variants. Among the five variants, *C* was very common and *D* was not detected in Romney sheep. In the Merino × Southdown-cross sheep both *B* and *C* were common. The differences in the frequency of the variants in these sheep breeds may suggests that this gene has been under different selection pressures and/or plays an important role in wool fibre development. Whether it reflects differences in wool fibre traits between the breeds, cannot be ascertained. 

While associations were detected between variation in *KRTAP21-2* and MSL, MFD, FDSD, PF and GFW, the most enduring association with *KRTAP21-2* variation appeared to for variation in MSL. For the fibre trait, there was a sizeable difference in the marginal mean for the common genotypes, and a difference that reinforced the conclusions drawn from the variant absence/presence models. 

Sheep with *E* had a lower MSL, and a trend for association with higher FDSD, MFD and PF. This is consistent with the correlation that has been reported between MSL and these other traits, with a moderate or close to moderate negative phenotypic correlation being reported between MSL; and FDSD (−0.35), MFD (−0.27) and PF (−0.24) [15]. The wool of higher MSL and lower FDSD, MFD and PF would be more desirable in the market, which means reducing the frequency of *E* on sheep group would fit the market demand. 

Wool from sheep with *A* have a higher GFW than those without *A*. Greasy fleece weight and CFW are reported to be strongly correlated at the phenotypic level (0.916) [15], yet the variant is not associated with variation in CFW. This suggests that *A* has an independent effect on GFW. The results from this study also suggest that nucleotide sequence variations in genes can have a functional effect, even if they are synonymous. Given that the only difference between variants *A* and *B* is a single synonymous nucleotide change, the difference in wool traits observed between genotypes *AB* and *BB* may be due to this variation. Although synonymous mutations do not change amino acid sequences, they may affect gene function in other ways. There is some evidence that they may change mRNA folding, translation and stability, protein folding, and miRNA-based regulation of expression [16]. This would be difficult to confirm without precise analysis of gene expression levels, and in the context of the small size and complexity of the wool follicle, that would not be an easy task. 

The possibility exists that the effects observed for *KRTAP21-2* may be due to its linkage to other *KRTAPs* on the same chromosome. Regardless of the potential for linkage, the extent of the genetic variation and its associations with MSL, suggests that ovine *KRTAP21-2* may have potential as a gene-marker for wool production. 

## Figures and Tables

**Figure 1 genes-11-00148-f001:**
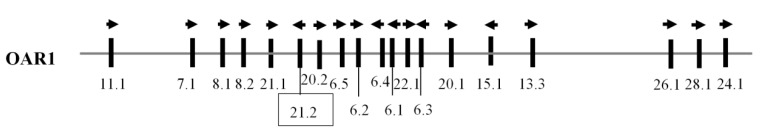
The chromosome 1 location of *KRTAPs*. The newly identified ovine *KRTAP21-2* is shown in a box, along with eighteen previously identified *KRTAPs*. Vertical bars represent the location of different *KRTAPs* and the arrowheads indicate the direction of transcription. The numbers below the bars indicate the name of the respective KAP genes (i.e., 24.1 is *KRTAP24-1*).

**Figure 2 genes-11-00148-f002:**
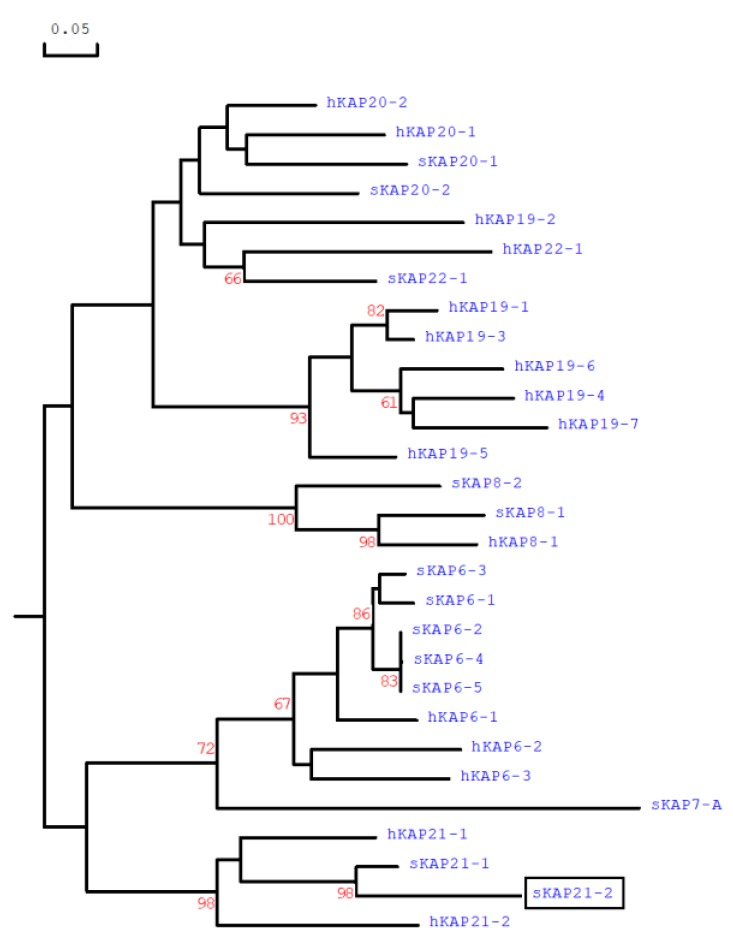
A phylogenetic tree positioning the newly identified ovine KAP21-2 (boxed) relative to selected HGT-KAPs from sheep (*Ovis aries*) and humans (*Homo sapiens*). The tree was constructed using predicted amino acid sequences for many of the proteins. The numbers at the forks of the tree indicate the bootstrap confidence values and only those equal to, or higher than 60%, are shown. The sheep KAPs are indicated with a prefix “s”, while the human sequences are indicated with “h”. The GenBank accession numbers for the ovine HGT-KAPs are: MH243552 and MH071391 (for sKAP20-1 and sKAP20-2 respectively), MF143980 for sKAP21-1 and KX377616 for sKAP22-1, and NM_001193399 (sKAP6-1), KT725832 (sKAP6-2), KT725837 (sKAP6-3), KT725840 (sKAP6-4), KT725845 (sKAP6-5), X05638 (sKAP7-1), X05639 (sKAP8-1), and KF220646 (sKAP8-2). The GenBank accession numbers for the human HGT-KAPs are: NM_181615 for hKAP20-1 and NM_181616 for hKAP20-2, NM_181619 for hKAP21-1 and NM_181617 for hKAP21-2, NM_181620 for hKAP22-1, NM_181602 (hKAP6-1), NM_181604 (hKAP6-2), NM_181605 (hKAP6-3), AJ457063 (hKAP7-1), AJ457064 (hKAP8-1), AJ457067 (hKAP19.1), NM_181608 (hKAP19-2), NM_181609 (hKAP19-3), NM_181610 (hKAP19-4), NM_181611 (hKAP19-5), NM_181612 (hKAP19-6), and NM_181614 (hKAP19-7).

**Figure 3 genes-11-00148-f003:**
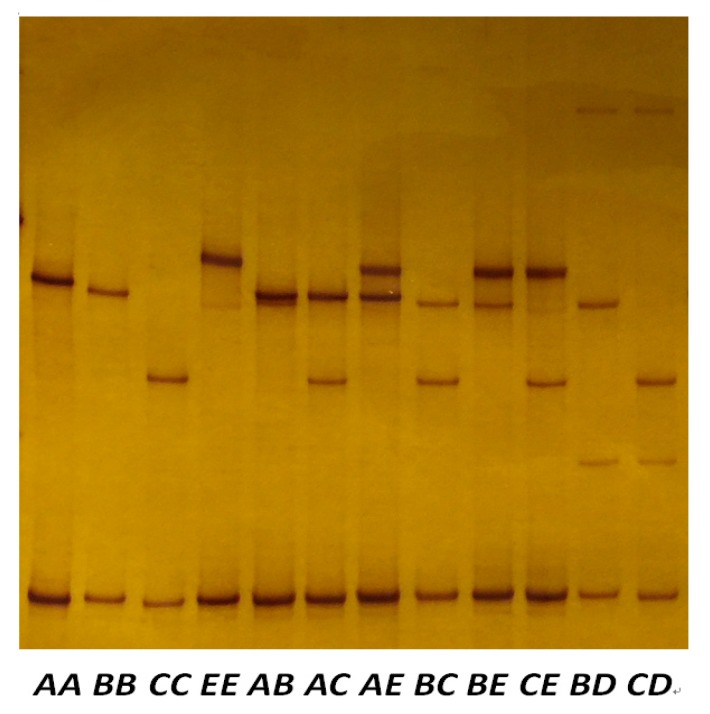
PCR-SSCP analysis of sheep *KRTAP21-2*. Different banding patterns came about from paring of five unique variant sequences (*A* to *E*) in either homozygous or heterozygous forms.

**Table 1 genes-11-00148-t001:** Sequence variation identified in ovine *KRTAP21-2*.

Nucleotide Position	Variant					Amino Acid Change
	*A*	*B*	*C*	*D*	*E*	
c.120	T	C	C	C	C	No change
c.144	T	T	T	C	T	No change
c.162	G	G	A	G	G	No change
c.166	G	G	G	A	A	p.V56I

**Table 2 genes-11-00148-t002:** Association between the absence or presence of *KRTAP21-2* variants and various wool traits (Mean ± SE) ^1^.

Trait ^2^	Variant Assessed	n		Single-Variant Model			Multi-Variant Model			
		Absent	Present	Absent	Present	*p*	Variants Fitted	Absent	Present	*p*
GFW	*A*	177	128	**2.3 ± 0.14**	**2.4 ± 0.14**	**0.030**	*E*	*2.2 ± 0.14*	*2.3 ± 0.14*	*0.078*
(kg)	*B*	145	160	2.3 ± 0.14	2.3 ± 0.14	0.739	*A,E*	2.3 ± 0.14	2.3 ± 0.14	0.913
	*C*	121	184	2.3 ± 0.14	2.3 ± 0.14	0.618	*A,E*	2.3 ± 0.14	2.3 ± 0.14	0.406
	*E*	252	53	2.3 ± 0.14	2.2 ± 0.15	0.173	*A*	2.3 ± 0.14	2.3 ± 0.15	0.590
CFW	*A*	177	128	1.8 ± 0.08	1.8 ± 0.09	0.474				
(kg)	*B*	145	160	1.8 ± 0.08	1.8 ± 0.09	0.645				
	*C*	121	184	1.7 ± 0.09	1.8 ± 0.08	0.442				
	*E*	252	53	1.8 ± 0.08	1.7 ± 0.09	0.228				
Yield	*A*	177	128	75.5 ± 1.46	74.8 ± 1.52	0.299				
(%)	*B*	145	160	75.5 ± 1.45	74.7 ± 1.53	0.244				
	*C*	121	184	74.7 ± 1.54	75.4 ± 1.45	0.305				
	*E*	252	53	75.4 ± 1.46	74.7 ± 1.65	0.504				
MSL (mm)	*A*	177	128	*80.9 ± 3.96*	*83.2 ± 4.05*	*0.092*	*E*	79.7 ± 3.95	80.6 ± 4.16	0.580
	*B*	145	160	81.8 ± 3.97	81.1 ± 4.08	0.619	*A,E*	80.2 ± 4.00	79.8 ± 4.16	0.728
	*C*	121	184	80.2 ± 4.09	82.0 ± 3.96	0.205	*A,E*	79.3 ± 4.08	80.8 ± 4.04	0.327
	*E*	252	53	**82.7 ± 3.91**	**77.0 ± 4.19**	**0.003**	*A*	**82.8 ± 3.92**	**77.6 ± 4.31**	**0.013**
MFD	*A*	177	128	19.7 ± 0.59	19.8 ± 0.60	0.628	*B,E*	19.7 ± 0.59	20.0 ± 0.62	0.358
(µm)	*B*	145	160	19.8 ± 0.58	19.5 ± 0.60	0.136	*E*	19.9 ± 0.59	19.6 ± 0.60	0.130
	*C*	121	184	19.7 ± 0.60	19.7 ± 0.59	0.946	*B,E*	19.9 ± 0.61	19.7 ± 0.60	0.462
	*E*	252	53	19.6 ± 0.58	20.1 ± 0.62	0.102	*B*	*19.5 ± 0.58*	*20.0 ± 0.62*	*0.098*
FDSD	*A*	177	128	4.1 ± 0.21	4.2 ± 0.21	0.549	*B,E*	4.2 ± 0.21	4.3 ± 0.22	0.260
(µm)	*B*	145	160	4.2 ± 0.20	4.1 ± 0.21	0.126	*E*	4.2 ± 0.21	4.1 ± 0.21	0.120
	*C*	121	184	4.2 ± 0.21	4.1 ± 0.21	0.787	*B,E*	4.2 ± 0.21	4.1 ± 0.21	0.330
	*E*	252	53	*4.1 ± 0.21*	*4.3 ± 0.22*	*0.068*	*B*	*4.1 ± 0.21*	*4.3 ± 0.22*	*0.065*
CVFD	*A*	177	128	21.0 ± 0.56	21.1 ± 0.59	0.593				
(%)	*B*	145	160	21.1 ± 0.56	20.9 ± 0.59	0.415				
	*C*	121	184	21.2 ± 0.59	21.0 ± 0.56	0.592				
	*E*	252	53	21.0 ± 0.56	21.3 ± 0.63	0.318				
MSS	*A*	177	128	21.9 ± 2.60	21.1 ± 2.67	0.427	*B,E*	21.5 ± 2.62	20.4 ± 2.78	0.262
(N/ktex)	*B*	145	160	*22.1 ± 2.59*	*20.5 ± 2.66*	*0.081*	*E*	*22.3 ± 2.62*	*20.7 ± 2.69*	*0.081*
	*C*	121	184	20.9 ± 2.68	21.9 ± 2.59	0.275	*B,E*	21.3 ± 2.70	21.6 ± 2.66	0.788
	*E*	252	53	*21.5 ± 22.60*	*22.1 ± 2.79*	*0.068*	*B*	21.2 ± 2.60	21.8 ± 2.78	0.630
MFC	*A*	177	128	96.1 ± 3.26	94.2 ± 3.46	0.281	*C*	95.2 ± 3.31	93.8 ± 3.46	0.435
(^o^/mm)	*B*	145	160	95.8 ± 3.29	95.0 ± 3.41	0.658	*C*	94.1 ± 3.44	95.3 ± 3.40	0.565
	*C*	121	184	*93.2 ± 3.49*	*96.2 ± 3.23*	*0.093*				
	*E*	252	53	94.9 ± 3.25	98.1 ± 3.83	0.218	*C*	93.9 ± 3.29	97.8 ± 3.82	0.126
PF	*A*	177	128	2.6 ± 0.74	2.7 ± 0.78	0.584	*E*	2.9 ± 0.75	3.5 ± 0.83	0.132
(%)	*B*	145	160	2.8 ± 0.75	2.4 ± 0.76	0.205	*E*	3.1 ± 0.77	2.7 ± 0.77	0.186
	*C*	121	184	2.6 ± 0.80	2.6 ± 0.74	0.905	*E*	2.8 ± 0.80	3.0 ± 0.76	0.639
	*E*	252	53	*2.4 ± 0.74*	*3.4 ± 0.84*	*0.055*				

^1^ Predicted means and standard error derived from GLMs with various factors being included into the models for different wool traits as described in the Materials and Methods. *p* < 0.05 are in bold, while 0.05 ≤ *p* < 0.10 are italicised; ^2^ GFW: Greasy fleece weight; CFW: Clean fleece weight; MFD: Mean fibre diameter; FDSD: Fibre diameter standard deviation; CVFD: Coefficient of variation of fibre diameter; MSL: Mean staple length; MSS: Mean staple strength; MFC: mean fibre curvature; PF: Picker factor (percentage of fibres over 30 microns).

**Table 3 genes-11-00148-t003:** The effect of *KRTAP21-2* genotype on various wool traits (Mean ± SE) ^1^.

Trait ^2^	*AC* (*n* = 64)	*AB* (*n* = 49)	*BB* (*n* = 22)	*BC* (*n* = 56)	*BE* (*n* = 27)	*CC* (*n* = 39)	*CE* (*n* = 21)	*p*
GFW (kg)	*2.3 ± 0.15^a^*	*2.3 ± 0.15^a^*	*2.0 ± 0.17^b^*	*2.3 ± 0.15^ab^*	*2.2 ± 0.16^ab^*	*2.2 ± 0.14^ab^*	*2.2 ± 0.16^ab^*	*0.054*
CFW (kg)	1.7 ± 0.12	1.7 ± 0.12	1.5 ± 0.14	1.7 ± 0.12	1.7 ± 0.13	1.7 ± 0.12	1.6 ± 0.14	0.184
Yield (%)	74.0 ± 2.10	73.0 ± 2.14	74.1 ± 2.40	73.9 ± 2.13	73.9 ± 2.26	76.4 ± 2.05	72.0 ± 2/37	0.138
MSL (mm)	**83.5 ± 4.23 ^a^**	**81.6 ± 4.32 ^ab^**	**78.8 ± 4.83 ^ab^**	**83.4 ± 4.28 ^ab^**	**77.2 ± 4.56 ^ab^**	**82.6 ± 4.12 ^ab^**	**74.2 ± 4.78 ^b^**	**0.026**
MFD (µm)	*19.7 ± 0.62 ^ab^*	*19.7 ± 0.6 3^ab^*	*19.6 ± 0.70 ^ab^*	*19.3 ± 0.62 ^b^*	*19.7 ± 0.66 ^ab^*	*19.4 ± 0.60 ^ab^*	*20.9 ± 0.70 ^a^*	*0.060*
FDSD (µm)	**4.1 ± 0.21 ^ab^**	**4.2 ± 0.22 ^ab^**	**4.1 ± 0.24 ^ab^**	**4.0 ± 0.22^b^**	**4.2 ± 0.23^ab^**	**4.1 ± 0.21^b^**	**4.6 ± 0.24^a^**	**0.019**
CVFD (%)	21.0 ± 0.82	21.1 ± 0.84	20.8 ± 0.94	20.4 ± 0.83	21.0 ± 0.88	20.8 ± 0.80	21.9 ± 0.93	0.345
MSS (N/ktex)	21.0 ± 2.77	18.7 ± 2.83	18.9 ± 3.16	20.5 ± 2.80	22.0 ± 2.98	22.6 ± 2.70	20.6 ± 3.13	0.259
MFC (^o^/mm)	89.2 ± 5.58	89.2 ± 5.69	87.8 ± 6.37	90.1 ± 5.65	88.9 ± 6.01	90.2 ± 5.43	96.5 ± 6.30	0.611
PF (%)	**2.6 ± 1.07 ^ab^**	**2.7 ± 1.09 ^ab^**	**2.2 ± 1.22 ^ab^**	**1.9 ± 1.09 ^b^**	**2.4 ± 1.16 ^ab^**	**2.1 ± 1.04 ^b^**	**4.7 ± 1.21 ^a^**	**0.041**

^1^ Predicted means and standard errors derived from GLMs with various factors being included into the models for different wool traits as described in the Materials and Methods. Means within rows that do not share a superscript letter (e.g. a or b) were different at *p* < 0.05. *p* < 0.05 are in bold, while 0.05 ≤ *p* < 0.10 are italicised; ^2^GFW: greasy fleece weight; CFW: clean fleece weight; MFD: mean fibre diameter; FDSD: fibre diameter standard deviation; CVFD: coefficient of variation of fibre diameter; MSL: mean staple length; MSS: mean staple strength; MFC: mean fibre Curvature; PF: prickle factor (percentage of fibres over 30 microns). Means within rows that do not share a superscript letter were different at *p* < 0.05 and the *p*-values bolded.

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
