# Peer review of "The Mean Staple Length of Wool Fibre Is Associated with Variation in the Ovine Keratin-Associated Protein 21-2 Gene"

_genes, 2020, doi:10.3390/genes11020148_

Round 1

Reviewer 1 Report

I haven’t additional suggestions regarding the proposed manuscripts.

Author Response

Thank you.

Reviewer 2 Report

The manuscript deals with the association of a candidate gene, KAP21-2, in a cluster of keratin genes in ovine and different wool quality traits. The work is  not very ambitious and uses an old albeit good technique to detect mutations which is SSCP technique.

I have some comments: First of all the paper lacks an objective and an hypothesis in the introduction even if it is a mere description of what happens with one particular keratin gene, I feel authors must justify why this gene and not another.

Also, this gene is situated in a cluster of keratin genes so, there would be of interest to compare it with the rest of sequences corresponding to the other genes and not to the human sequence. I can see that the same authors published in Animals an exact replicate of this text but referring to gene KAP21-1 instead of KAP21-2. Are the authors going to published each single study in different journals? I think one single paper dealing with the different keratin genes would be much more informative.

I suggest revising the English of the discussion mostly the first sentences.

In the references I can see that an important one is lacking: Ovine keratome: identification, localisation and genomic organisation of keratin and keratin-associated proteins. Yu et al Animal Genetics. 2018 Oct;49(5):361-370. doi: 10.1111/age.12694. Epub 2018 Jul 30.

Author Response

1. The manuscript deals with the association of a candidate gene, KAP21-2, in a cluster of keratin genes in ovine and different wool quality traits. The work is not very ambitious and uses an old albeit good technique to detect mutations which is SSCP technique.

Despite it being an old fashion technique, PCR-SSCP is reliable and cost-effective for genotyping a large number of samples. More importantly, it is a simple technique that is capable of detecting or differentiating alleles (i.e. SNP haplotypes/individual nucleotide changes).  

2. I have some comments: First of all the paper lacks an objective and an hypothesis in the introduction even if it is a mere description of what happens with one particular keratin gene, I feel authors must justify why this gene and not another.

We have added a brief description of why this gene was investigated.

3. Also, this gene is situated in a cluster of keratin genes so, there would be of interest to compare it with the rest of sequences corresponding to the other genes and not to the human sequence. I can see that the same authors published in Animals an exact replicate of this text but referring to gene KAP21-1 instead of KAP21-2. Are the authors going to publish each single study in different journals? I think one single paper dealing with the different keratin genes would be much more informative.

KRTAP21-2 belongs to the HGT group, so we compared this newly identified KRTAP21-2 with all of the HGT-KRTAPs that have been identified in sheep. Given that the KRTAPs sequences from sheep and human share a high sequence similarity, we have also included the human sequences in the phylogenetic tree.

We are looking at how to more effectively study the diversity within the keratin associated protein genes collectively, and also the diversity of the keratin genes, but these are large and diverse families of genes that are really only well annotated in humans. What is more the human genes are not that well characterized at the level of allelic/sequence variation, so in effect there is a lot to do (see below).

It would be great to have all of the KAP genes identified and the effects of these genes on wool traits published in a single paper. However, the identification of KAP genes, screening of sequence variation in these individual genes and the investigation of their associations with phenotypes would require a huge amount of work and time, and would probably take at least ten or twenty years to complete. Ovine KAP genes were first investigated about six decades ago and we have been actively working in this area for the last ten years, but so far only a small proportion (less than 30%) of KAP genes have been identified.        

4. I suggest revising the English of the discussion mostly the first sentences.

We have worked through the whole manuscript in its entirety to revise the English language.

5. In the references I can see that an important one is lacking: Ovine keratome: identification, localisation and genomic organisation of keratin and keratin-associated proteins. Yu et al Animal Genetics. 2018 Oct;49(5):361-370. doi: 10.1111/age.12694. Epub 2018 Jul 30.

Despite its fairly recent release, this paper needs to be reconciled with the other KAP literature. Specifically, it only identifies four KRTAP6 genes identified in the sheep genome, when five have been reported, and it appears to locate the two KRTAP21 genes within the KRTAP6 genes, which is different to what we have found. While bioinformatics approaches are providing really good information to resolving genome structures, they only provide part of the puzzle. In the context of the above comments about detailed analysis of ALL of the keratin and keratin-associated protein genes, then we believe another review of these genes is overdue, and given that we know Dr Yu, then we will endeavour to get his assistance if we take on this challenge.  
